# ID2SBVR: A Method for Extracting Business Vocabulary and Rules from an Informal Document

**Irene Tangkawarow** [1] , **Riyanarto Sarno** [2,*] **and Daniel Siahaan** [2]

1   Informatics Department, Faculty of Engineering, Universitas Negeri Manado, Minahasa 95618, Indonesia
2   Informatics Department, Faculty of Intelligent Electrical and Informatics Technology,
    Institut Teknologi Sepuluh Nopember, Surabaya 60111, Indonesia
*   Correspondence: riyanarto@if.its.ac.id; Tel.: +62-431-321847

**Abstract:** Semantics of Business Vocabulary and Rules (SBVR) is a standard that is applied in describing business knowledge in the form of controlled natural language. Business process designers develop SBVR from formal documents and later translate it into business process models. In many immature companies, these documents are often unavailable and could hinder resource efficiency efforts. This study introduced a novel approach called informal document to SBVR (ID2SBVR). This approach is used to extract operational rules of SBVR from informal documents. ID2SBVR mines fact type candidates using word patterns or extracting triplets (actor, action, and object) from sentences. A candidate fact type can be a complex, compound, or complex-compound sentence. ID2SBVR extracts fact types from candidate fact types and transforms them into a set of SBVR operational rules. The experimental results show that our approach can be used to generate the operational rules of SBVR from informal documents with an accuracy of 0.91. Moreover, ID2SBVR can also be used to extract fact types with an accuracy of 0.96. The unstructured data is successfully converted into semi-structured data for use in pre-processing. ID2SBVR allows the designer to automatically generate business process models from informal documents.

**Keywords:** SBVR; resource efficiency; fact type; operational rules; informal document to SBVR; natural language





## 1. Introduction

Business knowledge is an essential aspect of the early stages of systems development and evaluation in information systems engineering. The features of model-driven development and transformation are essential [1]. Determining business vocabulary and business rules is laborious because it requires much time and many resources. A more straightforward method to define business vocabulary (BV) and business rules (BR) is to conduct interviews and then extract them automatically using a natural language processing approach. The document resulting from the interview is an informal document.

The analysts who build a process model by gathering information require different techniques, such as document review and interviews [2]. Determining semantic business vocabulary and rules is challenging because of problems that result from the interview process. The processes described are not sequential, there are missing processes, and the interview document contains statements that are not relevant to the subject matter (noise). This documentation is not always well-structured and can be challenging to solve.

Informal documents usually refer to information that lacks a data model and that computer programs cannot easily use [3]. According to Praveen and Chandra [4] and Baig, Shuib, and Yadegaridehkordi [5], unstructured documents include files such as word processing documents, spreadsheets, PDFs, social media and message system content, graphical content, videos, and multimedia. The unstructured document does not have structured data information and precise data types and rules to apply to its stored data.

The difference between formal and informal documents is that formal documents are written following specific standards. In contrast, informal documents are more casual, conversational, and do not have a writing standard [6]. Examples of formal documents are documents containing standard operating procedures (SOP), laws and regulations, official script procedures, and policy documents. Examples of informal documents are news documents, documented interview results, memos, personal letters, and software requirements specifications (SRS).

The fact type defines the relationship between different concepts in BR and business process models: the noun concept indicates the name of the actor and the action verb indicates the process [7]. Informal documents that pass through the preprocessing stage are the basis for determining Semantics of Business Vocabulary and Business Rules (SBVR). SBVR is a standard to describe business knowledge in the form of controlled or structured natural language. Research to determine the transformation rules from SBVR to Business Process Modeling Notation (BPMN) in terms of structural rules and operational rules has been carried out. The research illustrated the transformation of data input using data already in the form of SBVR [8,9]. The enhanced Natural Language Processing (NLP) SBVR extraction provides recognition of entities, noun and verb phrases, and multiple associations [10]. They presented NLP-enhanced and pattern-based algorithms for SBVR automatic extraction from UML case diagrams. Previous research on NLP-enhanced algorithms was extended with a model-to-model (M2M) transformation approach [11]. According to Mishra and Sureka [12], there are inconsistencies between BPMN and SBVR. They generated Extensible Markup Language (XML) from a BPMN diagram, extracted triplets (actor, action, and object) using grammatical relations, searched node-induced sub-graphs, and applied algorithms to detect instances of semantic inconsistency. These indicate that recent research developments in natural language aim to deliver automatic model transformation.

In this paper, we present a novel approach to perform automatic translation of the informal document into fact type and operational rules of SBVR, called ID2SBVR. This method bridges the gap between an informal document, such as an open-ended interview, and a process model. We contribute to the model-driven information system development domain by automatic extraction of SBVR related to operational rules (behavioral rules) from informal documents. Specifically, the ID2SBVR searches for the sequence words, extracts the triplet, searches for the actor in a sentence, extracts the fact type, splits the fact type into compound, complex, or compound-complex sentences, and generates the operational rule of SBVR.

## 2. Related Works

The NLP research that has focused on business process modeling and SBVR has had various proposed methodologies. Several works that concern NLP, SBVR, and BPMN can be separated into six groups: works that discuss business process improvement and business process re-engineering to optimize the process and increase efficiency [13–16]; works that discuss SBVR transformation related to Software Requirements Specification (SRS) into XML [17–19]; works that discuss generating Unified Modeling Language (UML) class models from SRD using NLP [20]; works that discuss transformation from SBVR to BPMN where SBVR structured English (SE) specification is consistent and complete [12,21–24]; works that discuss producing SBVR from UML (use case diagram) [10,11]; and works that discuss generating natural language from business processes [2,25–27]. Further explanation regarding the grouping of related works is discussed below.

The Business Process Management (BPM) is an approach for advancing workflow in order to align processes with customer needs in an organization [13]. BPM covers both business process improvement and business process re-engineering [14]. Business Process (BP) focuses on re-engineering of processes and constant process improvement to achieve optimized procedures and increase efficiency and effectiveness [15].

Aiello et al. [17] investigated a mapping methodology and SBVR transformation grammar to produce rules that are ready to process in a rule engine. The main objective of their research is to overcome some weaknesses in the software development process that

can result in inconsistencies between the identification of domain requirements and the functionality of the software implemented. Arshad, Bajwa, and Kazmi [18] provided an approach for translating SBVR specifications of software requirements into an XML schema. The translation mapped verb concept, noun concept, characteristic, and quantification. Akhtar et al. [19] generated a knowledge graph based on Resource Description Framework SBVR (RDFS) from SBVR. They used SBVR rules and created a triplet (actor, action, and object), then generated the RDF and the RDFS [28].

Mohanan and Samuel [20] generated UML class models instantly from software requirement specifications (SRS) using a modern approach. Their approach used OpenNLP for lexical analysis and generated required POS tags from the requirement specification. In their further research, they developed a prototype tool that can generate accurate models in a shorter time [29]. It reduces the cost and budget for both the designers and the users.

BP modeling has a long-standing tradition in several domains. This discipline persists in the constant improvement of process and issue solving [21]. They examined the basic principle and the disparity between the specifications of BV and BR modeling and BP modeling. Another research transformed BR in SBVR into BPMN to assist the business expert in the requirement validation phase [22]. The focus was on the model transformation where the SBVR Structured English (SE) specification is consistent and complete. Kluza and Honkisz [24] presented an interoperability solution for transforming a subset of the SBVR rules into the BPMN and Decision Model and Notation (DMN) models. They combined process and decision models with translation algorithms to translate the SBVR vocabulary and structural and operational rules. Bazhenova, et al., [30] succeeded to identify a group of patterns that grab potential data representations in BPMN processes and it can be used to conduct the derivation of de-cision models related to current process models. Purificação and da Silva [31] succeeded in validating SBVR business rules that deliver content to assist users writing SBVR rules. This method supplied the functionality to update parts of the defined grammar with runtime and to locate and extract verb concepts that can be validated from the BR. Mishra and Sureka [12] investigated automatic techniques to detect inconsistencies between BPMN and SBVR. The research transformed rules to graphics and applied subgraph-isomorphism to detect instances of inconsistencies between BPMN and SBVR models.

Danenas et al. [10] succeeded in producing the SBVR from UML (use case diagrams) by automatic extraction. This research enhanced recognition of entities, entire nouns and verb phrases, improved various associations extraction capabilities, and produced better quality extraction results than their previous solution [11]. Their main contributions were pre- and post-processing algorithms and extraction algorithms using a custom-trained POS tagger.

Rodrigues, Azevedo, and Revoredo [25] investigated a language-independent framework for automatically generating natural language texts from business process models. They found empirical support that, in terms of knowledge representation, the textual work instructions can be considered equivalent to process models represented in BPMN. The research investigating the natural language structure showed that mapping rules and correlations between words representing the grammatical classes indicate a process element through keywords and/or verb tenses [2]. Furthermore, a semi-automatic approach successfully identified process elements from the natural language with a process description [26,27]. There were 32 mapping rules to recognize business process text elements using natural language processing techniques. This was discovered through an empirical study of texts comprising explanations of a process [2].

This current study presents two principal novel outcomes in terms of natural language processing and translating informal documents into SBVR. First, ID2SBVR generates the operational rules of SBVR from fact type, and, second, it can extract fact types from informal documents.

## 3. Materials and Methods

This section describe the research objectives of this study and the method of how ID2SBVR extracts operational rules of SBVR from informal documents. The method is explained in further detail in the following subsections.

### 3.1. Research Objectives

The main research objectives are: (i) to develop an implementation from concept to a fully functioning method to translate informal documents to SBVR; (ii) to analyze the correctness and accuracy of automatic fact type extraction by ID2SBVR; and (iii) to analyze the method's correctness and accuracy in generating operational rules of SBVR.

### 3.2. Mining Fact Type Candidate

An SBVR consists of business vocabulary (BV) and business rules (BR). BV is a vocabulary under business jurisdiction, and a BR is a rule under business jurisdiction [32]. A business vocabulary is composed of a noun concept, a verb concept, and an object type. In this work, we use a subset of the BV concept:

- The general concept is a noun concept. It is classified by its typical properties, e.g., noun person, noun place, noun thing;
- The verb concept can be auxiliary verbs, action verbs, or both.

BR signifies specific contexts of business logic; every BR is based on at least one fact type (FT). A combination of a verb concept and a noun concept is a fact type. The fact type determines the relationship between different BR concepts in BP models. The noun concept represents the actor, and the action verb concept represents a process. RuleSpeak in Object Management Group Annex-H [33] is an existing, well-documented BR notation developed by Business Rule Solutions (BRS). RuleSpeak is a set of rules for conveying business rules in a brief, business-friendly style [34]. It is not a language or syntax like structured English but rather a set of best practices for speakers of English.

BR in SBVR specifies two kinds of rules, structural and operational. Structural rules use such modal operators as necessary or possible/impossible. Operating rules use such modal operators as obligatory, permitted/forbidden [20,25]. The organizational settings are defined with rules of definition or structural rules in SBVR. For example, "**It is necessary that** <u>each customer</u> *has* <u>one customer ID</u>". The behavior of the noun person is defined with behavior rules or operation rules in SBVR. For example, '**It is obligatory that** <u>librarian</u> *sorts* <u>books</u> **after** <u>librarian</u> *receives* <u>books</u>'. Notation standard of SBVR as the controlled natural language used in this research [32]:

term of a noun concept that is part of used or defined vocabulary.
name for individual concepts and numerical values
verb for a fact type that is usually a verb or preposition, or both
keyword that accompanies designations or expressions; for example, obligatory, each, at most, at least, etc.

The SBVR notation that the ID2SBVR generates supports both SBVR structured English and RuleSpeak. Keyword modals available in BPMN include only the 'must' (or 'It is obligatory that') modal keyword because there is no way to present other modal operation in BPMN [20,27]. We executed part-of-speech (POS) and dependency parsing of all the example sentences in this paper using the Natural Language Processing (NLP) online software executor (https://corenlp.run; accessed on 12 April 2021) by the NLP Group of Stanford University (https://nlp.stanford.edu; accessed on 30 March 2021).

We consider only operational rules because they focus directly on the proper conduct of business activities, which in turn can be transformed into BPMN. To generate operational rule of SBVR, we perform several steps, and in the early stages, we search for a fact type candidate. The required fact type is one that has a noun, an active verb, and an object, which together form a triplet [12]. Sentence 1 and sentence 2 below illustrate fact type:

Sentence 1: 'The librarian analyzes the books needed'.

Sentence 2: 'Next, the librarian compiles the book using a tool'.

Sentence 1 and sentence 2 are examples of a simple sentence taken from an interview document. Figure 1a,b are the result from NLP that illustrated the POS from sentence 1 and sentence 2. The color and text in POS showed the label of POS tagging, where purple indicates DT label as determiner, blue indicates NN label as singular noun, blue indicates NNS label as plural noun, green indicates VBZ label as verb for third person singular in the present simple, green indicates VBN label as verb for past participle, green indicates VBG label as verb for gerund or present participle, and yellow indicates RB label as adverb. From the POS from both sentences, the fact type candidate is the arrangement of NN or NNS, VBZ, and objects consisting of arrays of NN or NNS, DT, VBN, and VBG. In the fact type rule, all DT and RB in the beginning of the sentence are not used. The fact type of sentence 1 and sentence 2 are:

'Librarian analyzes the books needed.'
'Librarian compiles the book using a tool.'

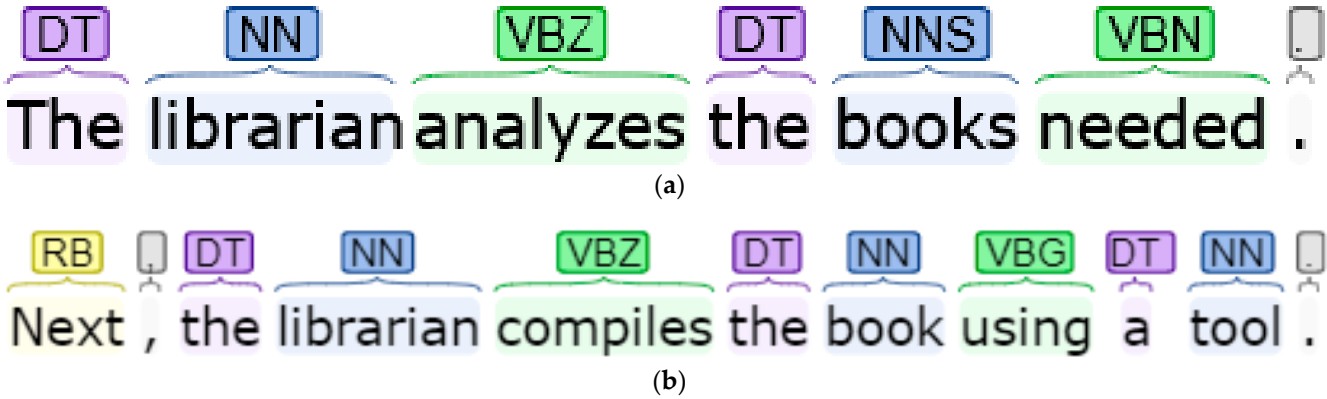

(a)

(b)

**Figure 1.** Part-of-speech of (**a**) sentence 1 and (**b**) sentence 2.

The BR structure as an operational rule of SBVR should be:

**'It is obligatory that** (fact type sentence 2) **after** (fact type sentence 1)'.

The operational rule of SBVR of sentence 1 and sentence 2 is:

**'It is obligatory that** librarian compiles the book using a tool **after** librarian analyzes the books needed'.

Based on the example above, the fact type consists of a noun (NN), a verb (VB/VBZ), and a noun as an object. In Figure 2 showing this research framework, the input document uses the interview document written in English. In the preprocessing step, the interview data is separated based on the interviewer's questions and interviewee's responses. The ID2SBVR searches the sentences for sequence words and the order between them. The ID2SBVR must indicate all the sequence words. Furthermore, the sentences that are not indicated have a sequence word parser to detect dependency and grammatical relations. The ID2SBVR searches the triplet (subject, active verb, and object) to fulfill the required fact type as operational rules. After that, the ID2SBVR determines the actor of each sentence with a noun. Then, the ID2SBVR extracts the fact type from fact type candidates. Next, ID2SBVR extracts the process name. Finally, the ID2SBVR generates the SBVR of operational rules.

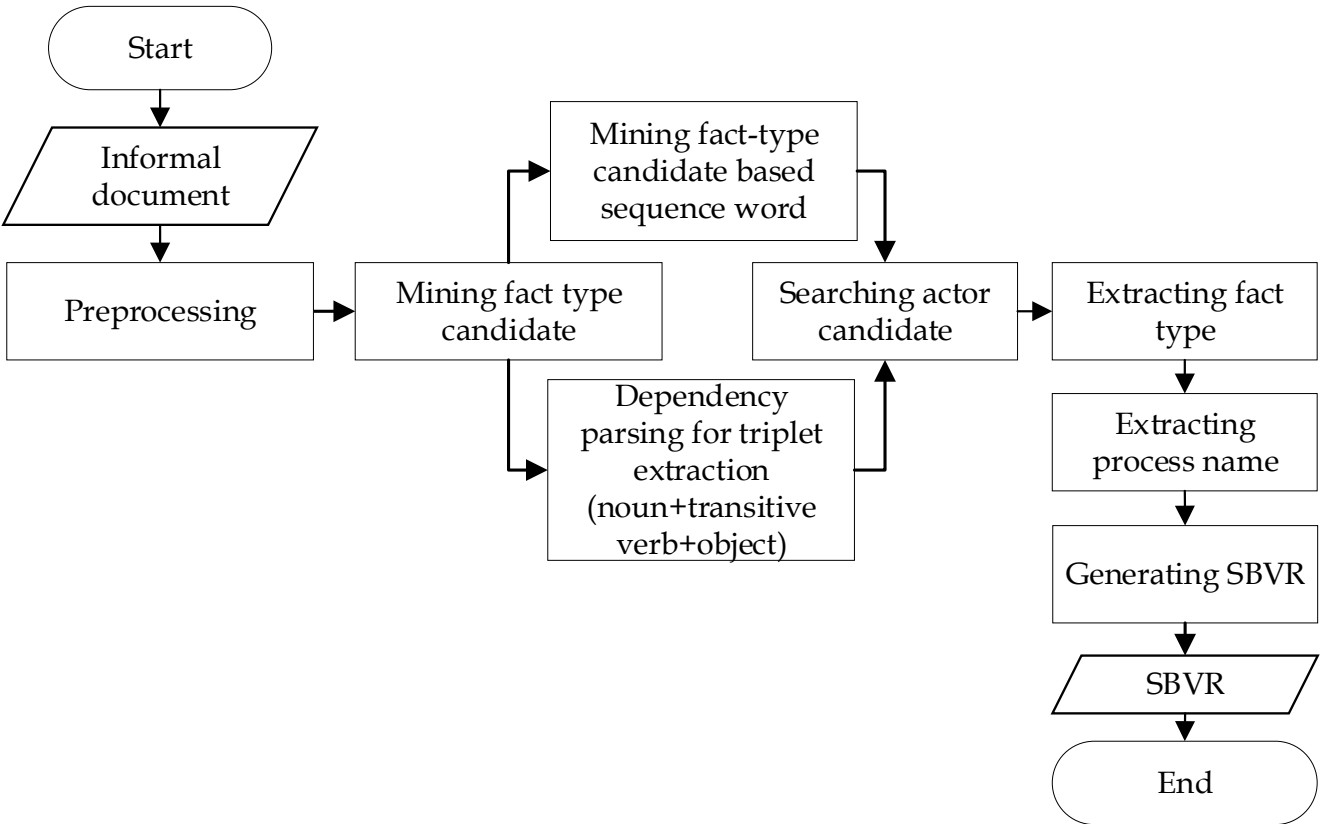

**Figure 2.** Research framework.

### 3.2.1. Mining the Fact Type Candidate Indicated by Sequence Words

Research on the classification of textual functions is primarily based on Hyland's model [35] of linguistic expressions in academic texts and semantic categories of linking adverbs [36]. There are five main functional semantic categories in Wang [37]: enumeration, code glosses, structuring signals, transition signals, and causal-conditional signals. Our research focused on enumeration especially for sequential words. Enumeration lists a series of steps, procedures, ideas, or subparts of the text, e.g., 'first', 'then', and 'finally', called sequence words.

At this stage, case folding is performed to change all letters in the document to lower case [38]. Any characters other than letters are removed, then tokenization is performed. Tokenization is the procedure of separating a text into words, phrases, or other meaningful parts, called tokens [38].

The ID2SBVR checks each sentence containing the sequence word. Sentences that have sequence words automatically become candidate fact types. The sequence words in question are: 'begins', 'starts', 'after', 'then', 'next', 'after that', 'when', 'finally', etc. The sentence with the sequence word in the interview is followed by an active sentence with a transitive verb. Furthermore, for sentences that do not have a sequence word, the next sentence is checked. If the following sentence has middle and ending sequence words, then the sentence is a fact type candidate. Middle and ending sequence words include 'after', 'then', 'next', 'after that', 'when', 'finally', etc. Algorithm 1 solves the problem of searching the fact type candidate based on sequence words.

### 3.2.2. Mining the Fact Type Candidate Indicated by Dependency Parsing

In this second phase of the solution, we used Python programming language with an open source library called the Natural Language Toolkit (NLTK) [39]. We consider only those rules which are action-oriented. The NLTK parse tree information represents the grammatical relation between words in the English structure. The result of dependency

parsing requires triplet extraction. Sentences with transitive verbs are needed to determine operational rules in BR.

The required nouns for this process are those related to the actor of the fact type. The next stage will be very dependent on the grammatical relation of the words in each sentence. The example of a simple sentence as a fact type candidate taken from an interview response in an informal document is 'The librarian analyzes the books needed'.

---

**Algorithm 1.** Fact type candidate based on sequence words

---

Data: input = answer
Output: fact type candidate
sequence word 1 = ['begins', 'starts', 'firstly', 'secondly', 'first', 'after', 'then', 'next', 'after', 'that', 'when', 'finally', 'furthermore', 'at the end']
sequence word 2 = ['after', 'then', 'next', 'after that'
data_clean = []
or answer in data_interview:
　　　change answer into lowercase
　　　change answer into token
　　　　　if answer is not in sequence word 1:
　　　　　　　if(index i is not = answer length—1):
　　　　　　　　　insert next sentence into data temporary
　　　　　　　　　change data temporary into token
　　　　　　　if answer is not in sequence word 2
　　　　　　　　　insert answer into fact type candidate
　　　　　end
　　　end
　　　end
　else:
　　　insert answer into fact type candidate
　end

---

The dependency parsing of the simple sentence in Figure 3 shows the structure of a simple sentence indicating fact type candidates. Figure 3 shows the structure of the sentence:

Noun (NN) as noun subject (nsubj):'librarian'
verb, 3rd person singular present simple (VBZ) as a transitive verb: 'analyzes'.
object consists of noun plural (NNS), verb past participle (VBN): 'the books needed'.

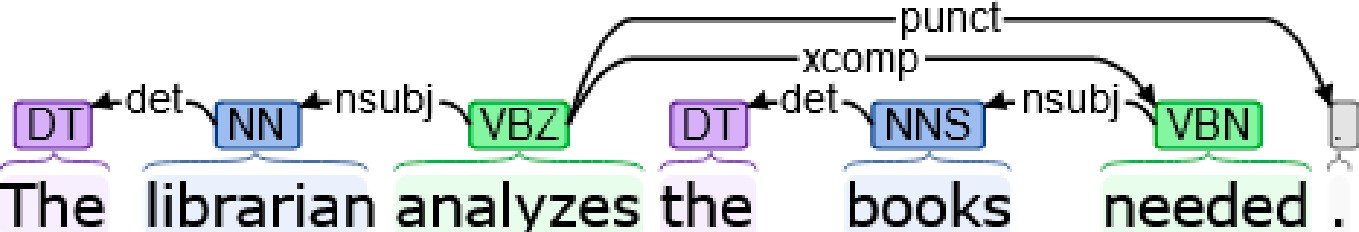

**Figure 3.** Dependency parsing of simple sentence.

The sentence structure consists of a noun (NN), a verb (VBZ), and an object.
The ID2SBVR determines the fact type candidate based on triplet extraction using Algorithm 2.

---

**Algorithm 2.** Fact type candidate based on triplet extraction

---

Data: input = answer
Output: Fact type candidate
for answer in enumerate (data_interview)
    if answer not in fact type candidate
    check noun subject in answer
    if noun subject exist in answer
        for index w in answer
        change answer to token
        if answer in tokens
        insert token with noun subject into data temporary
            insert token with verb into data temporary
            if token with verb = 'of': #*check temporary verb with of*
            for iteration as many as nlp
            check initialization= True
            if answer exist verb and answer is not exist 'of'
            check initialization= False
            if check = False:
                take index before the sentence
                insert sentence after index before the sentence
        end
        end
        end
        else:
        insert temporary subject
        insert temporary verb
        for iteration as many as nlp
        check initialization = True
        if answer verb followed by determiner (the) and object
        check initialization = False
        end
        else if answer verb followed by object
        check initialization = False
        if check initialization = False:
        take index before the sentence
        insert sentence after index before the sentence
        end
        end
      end
      end
  end
  end

---

### 3.3. Searching Actor Candidate

In this phase, we focus on searching for the actor candidate in the process. Actor candidate is a main actor of a fact type. The ID2SBVR searches the dependency parsing result that categorizes as nsubj (noun subject). We use the synset in WordNet NLTK corpus reader. Synset is a synonym set of words that share an ordinary meaning [39]. We use synset in searching all the noun subjects (nsubj) in the fact type candidate. Unfortunately, synset indicates each word as a noun, not a noun phrase. There are actors with a noun phrase, e.g., 'head of the library', 'the head of the employee subsection', and 'member candidate'. So, the ID2SBVR indicates the actor phrase using Algorithm 3.

---

**Algorithm 3.** Actor candidate

---

Data: data_interview
Output: actor candidate
for answer in enumerate (data_interview)
process answer to nlp
check result nlp process exist noun object
    if result nlp process exist noun object
    change answer to token
    for index w iteration as many answer exits noun subject
        if index w exist token
        insert token with noun subject into data temporary
           insert token with verb into data temporary
           if token with verb = 'of'
           for index i, t iteration as many nlp result
           check = True
           if answer exist verb and not an object
           if answer exist punct
           insert subject with answer
        end
        else:
           if the next word is followed by pucnt:
           insert answer to data temporary
        end
        else:
           insert answer with space to data temporary
           end
           end
        end
        if answer exist noun subject
        if previous word exist determiner
        insert previous word to data temporary
        else:
        insert subject into temporary subject
        if answer exist verb and answer is not exist 'of'
        check = False
        if check = False:
        if data temporary subject is not noun subject:
        show subject data temporary
        reset temporary
        break
        end
        end
        end
    end
    end
end

---

### 3.4. Extracting Fact Type

In this phase, we focus on extracting the complex fact type into a fact type. A sentence as a fact type has a noun, an active verb, and an object (see Section 3.1). Mishra and Sureka [12] named it a "triplet" (noun, verb, object). A sentence with more than one fact type categorizes as a compound sentence, a complex sentence, or a compound-complex sentence.

A compound sentence has a coordinating conjunction (CC) that joins two independent clauses, e.g., 'for', 'and', 'nor', 'but', 'or', 'yet', and 'so' [40]. Except for very short sentences, a comma (,) appears right before the coordinating conjunction. The example of a compound sentence in Figure 4 shows the CC 'and' join two independent clauses.

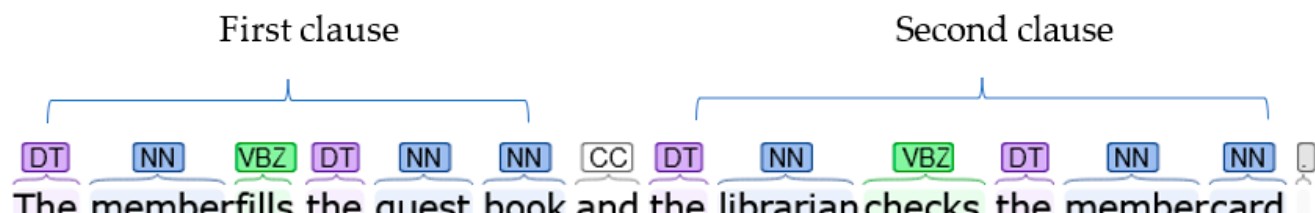

**Figure 4.** A compound sentence.

Based on the fact type structure, the compound sentence in Figure 4 should split into two fact types as two independent clauses. The first fact type is the clause before the CC, and the second fact type is the clause after the CC. The other step in splitting the compound sentence is identifying the fact type with no noun as a noun subject.

The second clause after the CC starts with the verb 'writes.' The noun subject should be 'librarian' as the first clause. The ID2SBVR adjusts it with the noun subject of the first fact type. Algorithm 4 shows the procedure for splitting the compound sentence into simple sentences as fact type. There are seven coordinating conjunctions listed. The fact types of Figure 4 are:

Fact type 1: 'Member fills the guest book'.
Fact type 2: 'Librarian checks the member card'.

---

**Algorithm 4.** Fact type from compound sentence

Data: Fact type candidate
Output: Fact type
sentence=compound sentence
split data (r 'and | for | nor | but | or | yet | so', sentence)
show data

---

A complex sentence has one independent clause and one or two dependent clauses [40]. It always has a subordinating conjunction ('because', 'since', 'after', 'although', 'when') or a pronoun, such as 'who', 'which', and 'that'.

Based on the fact type structure, a complex sentence in Figure 5 should split into two fact types. The first fact type is the clause before the subordinating conjunction 'before', and the second fact type is the clause after the subordinating conjunction. If there is no noun subject in the double fact type (after the subordinating conjunction or comma ','), the ID2SBVR adjusts it with the first fact type's noun subject. Algorithm 5 shows the algorithm for splitting the complex sentence into a simple sentence as fact type. All the subordinating conjunctions are listed to make the splitting easy.

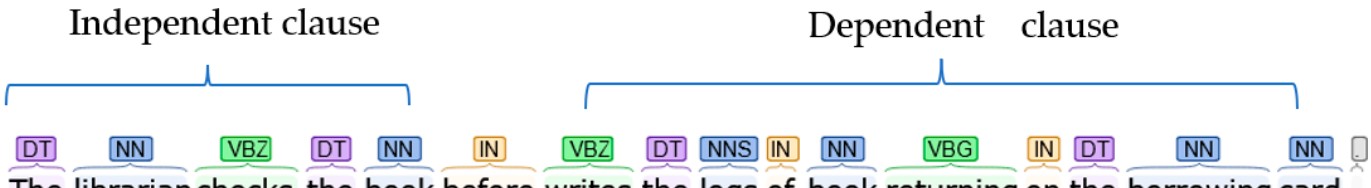

**Figure 5.** The example of a complex sentence.

The fact types of Figure 5 are:

Fact type 1: 'Librarian checks the book'.
Fact type 2: 'Librarian writes the logs of returning book on the borrowing card'.

A compound-complex sentence is a sentence with a combination of a compound sentence and a complex sentence [40]. A compound-complex has three or more clauses and at minimum has two independent clauses and one dependent clause.

Based on the fact type structure, a compound-complex sentence in Figure 6 should split into 3 fact types. The first fact type is the clause before the subordinating conjunction 'then', the second fact type is the clause after the subordinating conjunction, and the third fact type is the clause after the CC 'and'. Algorithm 6 shows the process to split a compound-complex sentence. All the coordinating conjunction words (CC) and subordinating conjunction words are listed in the Algorithm 6.

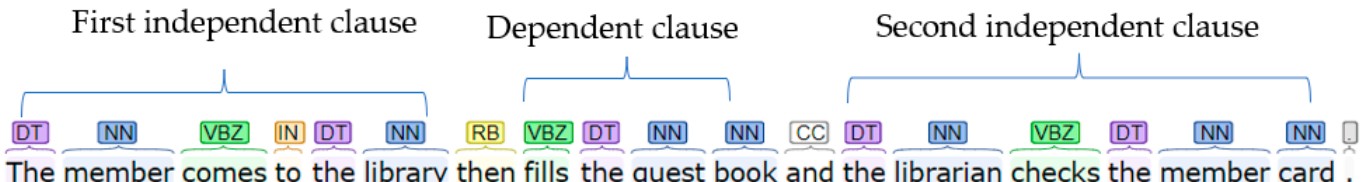

**Figure 6.** The example of compound-complex sentence.

The fact types of Figure 6 are:

Fact type 1: 'Member comes to the library'.
Fact type 2: 'Member fills the guest book'.
Fact type 3: 'Librarian checks the member card'.

---

**Algorithm 5.** Fact type from compound-complex sentence

---

Data: Fact type candidate
Output: Fact type
sentence = 'compound-complex sentence'
Split data (r 'and | for | nor | but | or | yet | so | after | once | until | although | then | provided that | when | as | rather than | whenever | because | since | where | before | so that | whereas | even if | than | wherever | even though | that | whether | if | though | while | in order that | unless | why', sentence)
Show data

---

### 3.5. Extracting Fact Type

At this stage, the ID2SBVR extracts the process name and separates each fact type based on the name of the extracted process. The process name comes from the list of questions in the data. The process name is important for the transformation of SBVR into BPMN. BMPN will use it as the pool name. Algorithm 7 extracts the process name from the interview questions in the data.

---

**Algorithm 6.** Extracting process name

---

Data: Input = Question
Output: Process name
    If previous question is not question data:
    Show new line
    show question
    process question to nlp process
        for index t iteration as many result of nlp process
            if question exist compound type
                if previous word of question exist amod type
                insert result of nlp process with amod type to

---

| **Algorithm 6.** *Cont.* |
|---|
|           temporary data<br>          for iteration compound type until the last word<br>          if result of nlp exist punct type<br>          insert result of nlp process to temporary data<br>          show temporary data<br>          end<br>        end<br>      end<br>  end<br>  end |

*3.6. Generating SBVR*

In this phase, we focus on the RuleSpeak SBVR for operational rules generated from the fact type. We are concerned with the operational rules in SBVR because our future research is to transform the operational rules into BPMN. The ID2SBVR uses the keywords or RuleSpeak SBVR with:

'It is obligatory that <fact type2> after <fact type1>'.

Algorithm 7 generates a fact type into an operational rule in SBVR. The ID2SBVR must consider the order of the fact type.

| **Algorithm 7.** Fact type operational rule in SBVR |
|---|
| Data: Fact type candidate<br>Output: SBVR<br>#prosessbvr<br>topic = 0<br>sbvr S = result of sbvr process interview data<br>sbvr C = result of sbvr process check sentence<br>for index i iteration as many as result of sbvr process<br>      if index i = 0 or sbvr is not exist previous sbvr<br>          show topic<br>          end<br>          if index i is not exist complete sentences and sbvr S exist next<br>          sbvr S and sbvr C is 0 and next sbvr C is 0<br>          print ('It is obligatory that', next complete<br>          sentence, 'after', complete sentence)<br>      end<br>  end |

Based on the extracting fact type in the previous phase, we generate the fact type operational rule in RuleSpeak SBVR. The example of the SBVR follows.

**'It is obligatory that** member candidates fill and complete the form **after** librarian submits it to the circulation or reference sub—librarian section afterwards'.

**'It is obligatory that** member candidates enclose tuition fee slip and two pass member candidates photos **after** member candidates fill and complete the form'.

3.6.1. Conjunction

Conjunction is a binary logical operation that formulates that the meaning of each of its logical operands is true. The example of a conjunction appears below. Response 2 shows the compound sentence as a conjunction. In response 2 with conjunction, we separate the noun subject, verb, and object.

Response 1: 'Firstly, the librarian determines the exhibition themes'.
Response 2 with conjunction: 'The librarian selects material, librarian determines design, librarian prepares support event, and librarian prepares promotion concept'.

Response 3: 'Then, the librarian does the exhibition together with the team'.

The ID2SBVR generates the fact type based on the response above. Fact type 1 generates from response 1, fact type 2 generates from response 2 with conjunction, and fact type 3 generates from response 3.

Fact type 1: 'librarian determines the exhibition themes'.
Fact type 2 with conjunction: 'librarian selects materials, librarian determines design, librarian supports event, and librarian prepares promotion concept'.
Fact type 3: 'librarian does the exhibition together with the team'.

The ID2SBVR generates the SBVR 1 and SBVR 2 based on the fact type 1, fact type 2 with conjunction, and fact type 3.
The composition of SBVR 1:
<**It is obligatory that**> fact type 2 with conjunction separated with 'and' <**after**> fact type 1.
The composition of SBVR 2:
<**It is obligatory that**> fact type 3 <**after**> fact type 2 with conjunction separated with 'and'.
SBVR 1:
'**It is obligatory that** librarian selects materials **and** librarian determines design **and** librarian supports event **and** librarian prepares promotion concept **after** librarian determines the exhibition themes.'
SBVR 2:
'**It is obligatory that** librarian does the exhibition together with the team **after** librarian selects materials **and** librarian determines design **and** librarian supports event **and** librarian prepares promotion concept.'

### 3.6.2. Exclusive Disjunction

Exclusive disjunction in SBVR is a binary logical operation that indicates that the meaning of one logical operand is true and the meaning of the other logical operand is false [32]. Fact type with exclusive disjunction:

fact type: 'member active registered'
fact type: 'member allowed to enter the library **else** not allowed to enter'.

SBVR:
'**It is obligatory that** member allowed to enter the library **after** member active registered **else** not allowed to enter'.

### 3.6.3. Inclusive Disjunction

Inclusive disjunction is a binary logical operation that indicates that the meaning of at least one of its logical operands is true [32].
Response 1 with disjunction:
'The librarian categorizes scientific paper as thesis, librarian categorizes scientific paper as dissertation, or librarian categorizes other scientific paper'.
Response 2:
'Then, librarian puts the scientific paper in the cabinet'.
The ID2SBVR generates fact type 1 from response 1 and fact type 2 from response 2.
Fact type 1 with inclusive disjunction:

'librarian categorizes scientific paper as thesis'
'librarian categorizes scientific paper as dissertation'
'librarian categorizes other scientific paper'

Fact type 2:

'librarian puts the scientific paper in the cabinet.'

SBVR:
'**It is obligatory that** librarian puts the scientific paper in the cabinet **after** librarian categorizes scientific paper as thesis **or** librarian categorizes scientific paper as dissertation **or** librarian categorizes other scientific paper'.

## 4. Results and Discussion

This chapter describes the scenario of the experiment, a case study in a university library, and the results and discussion of each stage in this research.

### 4.1. Scenario

The experiment was performed to answer the basic question, "What is the correctness and accuracy of automatic extracting of fact type and generating SBVR from interview response as an informal document when compared to the benchmark result provided by manual extraction?" An evaluation was carried out to measure the correctness and accuracy of the information retrieval (IR) produced by the ID2SBVR.

The confusion matrix is used to measure the classification method's performance, in this case, precision, recall or sensitivity, specificity and accuracy. In basic terms, the confusion matrix contains information that compares the system classification results carried out with the expected results [41]. We defined the confusion matrix rule, i.e., the data extracted true as true positive (TP); data extracted false as false positive (FP); deviation between data extracted false and benchmark as false negative (FN); and deviation between data extracted true and benchmark as true negative (TN).

The IR evaluation metrics commonly used in text classification are precision, recall, fscore, and accuracy [42]. We evaluated the ID2SBVR developed to extract operational rules in terms of precision, recall, specificity, and accuracy. Precision is the ratio of correctly identified fact type to the total number of fact type extracted; the recall is a ratio of correctly determined fact type to several suitable fact types; accuracy is the accurate prediction ratio (positive and negative) to the overall data. We need to measure the precision and recall rather than accuracy because, with accuracy, the results do not necessarily match the necessary data [43]. The research of Skersys, Danenas, and Butleris [11] measures the accuracy of automatic and wizard-assisted semi-automatic extraction of SBVR from UML.

The proposed approach for extracting the fact type and generating SBVR were evaluated using one scenario. The scenario of this experiment involves the list of procedures in the library. The evaluation phase consists of the procurement process, inventory process, processing section, member registration, book borrowing process, book returning process, reference sub-section process, librarian promotion process, and the process to review the promotion document. There are six steps of the evaluation to extract the fact type and generate SBVR, i.e., (1) build a ground truth from fact types extracted by domain expert; (2) build a ground truth from SBVR extracted by domain expert; (3) measure the performance of ID2SBVR in extracting fact types using precision, recall, specificity and accuracy; (4) measure the accuracy of ID2SBVR in generating SBVR.

To build the ground truth in steps (1) and (2), we involved domain experts in the field of business process modeling. Domain experts manually executed the complete set of all processes in the library. The domain expert determines all the fact types of the semi-structured interview document. After that, the domain experts compile fact types into SBVR. The result of fact types and SBVR from domain experts can directly be used to evaluate the ID2SBVR method.

### 4.2. Description of University Library Case Study

This experiment uses a case study consisting of informal data as the interview response written in standard English. The interviews took place in two university libraries. The first university library has 24,354 book titles with 71,368 copies. This library has 12 librarians and 14 staff. The second university library has 145,252 book titles with 210,605 copies in its database. Of these, in its physical collection, this library has 89,672 book titles with 138,391 copies. The working staff consists of 27 librarians and 24 administrative staff and officials.

In the pre-processing phase, we divided all the questions and responses into columns. We also split each response per sentence into rows. Every question has more than one response. Each row response represents one sentence. The first dataset contains 1247 words,

forming 110 sentences containing 61 sequence words. The details of the dataset are shown in Table 1 and the complete dataset has been published in online repository as dataset_university (https://doi.org/10.6084/m9.figshare.15123879.v1). The other dataset, dataset_university (https://doi.org/10.6084/m9.figshare.15123972.v2), contains 2585 words across 288 sentences containing 195 sequence words. The summary details of the dataset are shown in Table 2.

**Table 1.** Dataset_university1.

| Question | Process | Number of | | | | | | Sentence |
|---|---|---|---|---|---|---|---|---|
| ID | Name | Sentence | Word Response | Verb | Sequence Word | Compound | Complex | Compound-Complex |
| q1 | - | 2 | 24 | 3 | - | - | - | - |
| q2 | procurement section | 12 | 146 | 27 | 6 | 3 | 1 | - |
| q3 | processing section | 6 | 74 | 6 | 5 | 1 | 2 | - |
| . . . | | . . . | . . . | . . . | . . . | . . . | . . . | . . . |
| q16 | inventarisation | 6 | 61 | 5 | 4 | - | - | 1 |
| Total | | 110 | 1247 | 171 | 61 | 11 | 10 | 7 |

**Table 2.** Dataset_university2.

| Question | Process | Number of | | | | | | Sentence |
|---|---|---|---|---|---|---|---|---|
| ID | Name | Sentence | Word Response | Verb | Sequence Word | Compound | Complex | Compound-Complex |
| r1 | - | 4 | 63 | 6 | - | 5 | - | - |
| r2 | LPBP Lelang | 8 | 60 | 8 | 7 | - | - | - |
| r3 | LPENGOLBP book | 7 | 51 | 7 | 6 | - | - | - |
| . . . | | . . . | . . . | . . . | . . . | . . . | . . . | . . . |
| r33 | Training, seminars, and workshops held in library | 18 | 264 | 17 | 6 | 3 | 1 | - |
| Total | | 288 | 2585 | 337 | 195 | 17 | 13 | 16 |

There are no specific rules in the interview for determining the sequence of responses. However, the SBVR operational rule requires a sequence of responses indicated as a procedure. The ID2SBVR needs to identify the sequence words of each response to represent the sequence of the fact type. Furthermore, not every response has a sequence word, but it may have a candidate fact type. The ID2SBVR indicates the transitive verb in those responses with the extracted triplet (see Section 3.2.1). The sequence word identifies fact types and the order between fact types, not the sequence in SBVR.

### 4.3. Extracting Fact Type

In this experiment, the accuracy of the ID2SBVR-extracted fact type shows an average of 0.98. Table 3 presents the final calculation of the precision, recall, and accuracy collected from automatic extracting of fact type using dataset_university1. In the experimental results, the overall average values obtained are 0.98 in terms of precision, recall or sensitivity, specificity, and accuracy.

**Table 3.** Experimental results of extracting fact type in dataset_university1.

| Question ID | Precision | Recall | Specificity | Accuracy |
|---|---|---|---|---|
| q1 | - | - | | - |
| q2 | 1.00 | 1.00 | 1.00 | 1.00 |
| q3 | 0.83 | 0.83 | 0.83 | 0.83 |
| q4 | - | - | | - |
| q5 | 1.00 | 1.00 | 1.00 | 1.00 |
| q6 | 0.88 | 0.88 | 0.88 | 0.88 |
| q7 | 1.00 | 1.00 | 1.00 | 1.00 |
| q8 | 1.00 | 1.00 | 1.00 | 1.00 |
| q9 | 1.00 | 1.00 | 1.00 | 1.00 |
| q10 | 1.00 | 1.00 | 1.00 | 1.00 |
| q11 | 1.00 | 1.00 | 1.00 | 1.00 |
| q12 | 1.00 | 1.00 | 1.00 | 1.00 |
| q13 | 1.00 | 1.00 | 1.00 | 1.00 |
| q14 | 1.00 | 1.00 | 1.00 | 1.00 |
| q15 | 1.00 | 1.00 | 1.00 | 1.00 |
| q16 | 1.00 | 1.00 | 1.00 | 1.00 |
| Average | 0.98 | 0.98 | 0.98 | 0.98 |

The maximum accuracy value is 1.00 because the fact type is contracted according to the fact type benchmark. The q3 has minimum accuracy value of 0.83 because in q3 there are two fact types detected incorrectly from a total of seven fact types. An error in q3 occurs where the detected fact type combines two fact types with the conjunction 'or'.

fact type: librarian submits into the circulation or librarian submits into reference sub section afterwards.

The results of the fact type extraction should occur by splitting the sentence:

fact type: librarian submits into the circulation.
fact type: librarian submits into reference sub section afterwards.

Furthermore, the ID2SBVR accuracy in extracted fact type using dataset_university2 shows an average value of 0.93. Table 4 presents the precision, recall, and accuracy of ID2SBVR using dataset_university2. In the experimental results, the overall average values obtained are precision 0.98, recall 0.91, specificity 0.98, and accuracy 0.95.

**Table 4.** Experimental results of extracting fact type in dataset_university2.

| Question ID | Precision | Recall | Specificity | Accuracy |
|---|---|---|---|---|
| r1 | - | - | - | - |
| r2 | 1.00 | 1.00 | 1.00 | 1.00 |
| r3 | 1.00 | 1.00 | 1.00 | 1.00 |
| r4 | 1.00 | 1.00 | 1.00 | 1.00 |
| r5 | 1.00 | 1.00 | 1.00 | 1.00 |
| r6 | 1.00 | 1.00 | 1.00 | 1.00 |
| r7 | 1.00 | 1.00 | 1.00 | 1.00 |
| r8 | 1.00 | 1.00 | 1.00 | 1.00 |
| r9 | 1.00 | 1.00 | 1.00 | 1.00 |
| r10 | 1.00 | 1.00 | 1.00 | 1.00 |
| r11 | 1.00 | 1.00 | 1.00 | 1.00 |
| r12 | 0.86 | 0.75 | 0.88 | 0.81 |
| r13 | 1.00 | 0.80 | 1.00 | 0.90 |
| r14 | 1.00 | 0.86 | 1.00 | 0.93 |
| r15 | 1.00 | 1.00 | 1.00 | 1.00 |
| r16 | 1.00 | 0.88 | 1.00 | 0.94 |

**Table 4.** *Cont.*

| Question ID | Precision | Recall | Specificity | Accuracy |
|---|---|---|---|---|
| r17 | 1.00 | 1.00 | 1.00 | 1.00 |
| r18 | 1.00 | 1.00 | 1.00 | 1.00 |
| r19 | 0.80 | 0.67 | 0.83 | 0.75 |
| r20 | 1.00 | 1.00 | 1.00 | 1.00 |
| r21 | 1.00 | 0.89 | 1.00 | 0.94 |
| r22 | 1.00 | 1.00 | 1.00 | 1.00 |
| r23 | 1.00 | 1.00 | 1.00 | 1.00 |
| r24 | 1.00 | 0.81 | 1.00 | 0.91 |
| r25 | 1.00 | 0.83 | 1.00 | 0.92 |
| r26 | 1.00 | 1.00 | 1.00 | 1.00 |
| r27 | 0.87 | 0.76 | 0.88 | 0.82 |
| r28 | 0.83 | 0.63 | 0.88 | 0.75 |
| r29 | 1.00 | 1.00 | 1.00 | 1.00 |
| r30 | 0.95 | 0.83 | 0.96 | 0.89 |
| r31 | 1.00 | 0.77 | 1.00 | 0.88 |
| r32 | 1.00 | 0.67 | 1.00 | 0.83 |
| r33 | 1.00 | 1.00 | 1.00 | 1.00 |
| Average | 0.96 | 0.89 | 0.98 | 0.93 |

The maximum accuracy values occur in r2–r11, r15, r18, r20, r22, r23, r26, r29, and r33 because the fact type is calculated according to the fact type benchmark. The r19 and r28 data points display a minimum accuracy value of 0.75. In fact, in r19, two fact types were detected incorrectly and two fact types were missing. The incorrectly detected fact types indicate that there is no subject in the sentence and the missing fact types indicate that the sentence's verb is not recognized by ID2SBVR. The two missing sentences are:

'First, member hands over the book and receipt'.
'Next, staff files borrowing receipts to its shelf'.

The results of the fact type extraction should be:

'member hands over the book and receipt'.
'staff files borrowing receipts to its shelf'.

The other minimum accuracy value occurs in r28 because there are two missing fact types. As previously, the missing fact types indicate that the verb and subject of the sentence are not recognized by ID2SBVR. The two sentences that are missing are:

'then, student scans the id card barcode'.
'if student chooses to save the file, then the file will be saved on the storage device, else prints the file'.

The results of the fact type extraction should be:

fact type: 'student scans the id card barcode'.
fact type: 'student chooses to save the file'.
fact type: 'the file will be saved on the storage device'.
fact type: 'student prints the file'.

In comparison with the study by Lopez et al. [2], their prototype for the extraction of business process elements in natural language text showed precision, recall, and accuracy values of 0.92, 0.84, and 0.88, respectively, based on a set of 56 texts. The advantage of ID2SBVR is that natural language is first extracted to SBVR which becomes standard English. This facilitates the transformation of the SBVR into BPMN. The research by Arshad et al. [18] used an approach that transformed SBVR to XML; this approach displayed an average recall value of 0.89 while the average precision value was 0.96.

Based on the results of performance calculations shows in Figure 7, from both datasets, analysis of the university1 dataset yielded higher precision, recall, and accuracy values than university2, while both datasets generated the same specificity value. The number

of different data points in the dataset does not significantly affect the performance of this method. However, grammatical errors will increase false positives (FP) and false negatives (FN) which markedly affects the accuracy of this method.

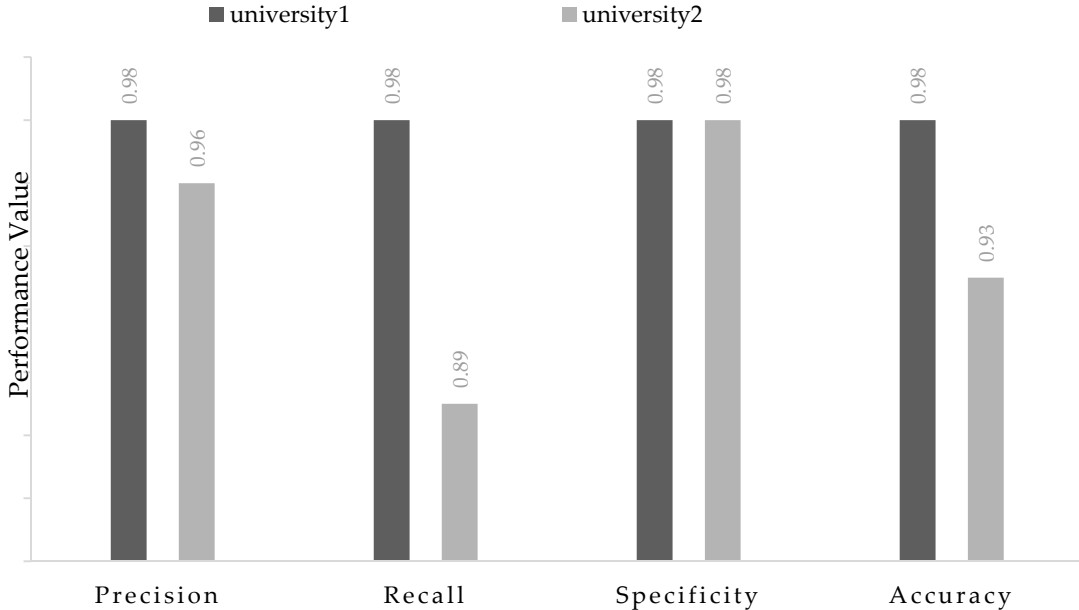

**Figure 7.** Graphical representations of precision, recall, specificity, and accuracy of ID2SBVR-extracted fact type.

*4.4. Generating SBVR*

The total operational rules generated by ID2SBVR was 103 operational rules for dataset_university1 and 209 operational rules for dataset_university2. Accuracy in generating SBVR is calculated for each process by comparing the number of SBVR detected correctly with the total SBVR generated. In dataset_university1, 99 SBVR were was correctly detected (96.2% of the total SBVR), while there were four SBVR with errors (3.8% of the total SBVR). The average accuracy value for generating SBVR using ID2SBVR was 0.94, as shown in Table 5. For the processed q3 and q6, there were two SBVR errors generated from a total of five SBVR. The errors in q3 and q6 were both caused by incorrect fact types. In terms of the variance of the ID2SBVR accuracy data, in dataset_university1 the value was 0.02 with a standard deviation [44] of 0.15.

**Table 5.** Experimental results of generating SBVR in dataset_university1.

| Question ID | SBVR | Error Due to Wrong Fact Type | Error Due to Wrong Sequencing | Accuracy |
|---|---|---|---|---|
| q1 | - | - | - | - |
| q2 | 11 | - | - | 1.00 |
| q3 | 5 | 2 | - | 0.60 |
| q4 | - | - | - | - |
| q5 | 3 | - | - | 1.00 |
| q6 | 5 | 2 | - | 0.60 |
| q7 | 4 | - | - | 1.00 |
| q8 | 4 | - | - | 1.00 |
| q9 | 6 | - | - | 1.00 |
| q10 | 8 | - | - | 1.00 |
| q11 | 5 | - | - | 1.00 |
| q12 | 8 | - | - | 1.00 |
| q13 | 10 | - | - | 1.00 |
| q14 | 13 | - | - | 1.00 |

**Table 5.** *Cont.*

| Question ID | SBVR | Error Due to Wrong Fact Type | Error Due to Wrong Sequencing | Accuracy |
|---|---|---|---|---|
| q15 | 15 | - | - | 1.00 |
| q16 | 11 | - | - | 1.00 |
| | | | Average | 0.94 |
| | | | Variance | 0.02 |
| | | | Standard Deviation | 0.15 |

In the dataset_university2, SBVR correctly detected 186 SBVR or 89% of the total SBVR, with an 11% error rate. The errors were caused by wrong fact types or missing fact types causing wrong sequencing. The average accuracy of ID2SBVR-generated SBVR was 0.88, as shown in Table 6. The statistical analysis of the ID2SBVR accuracy values in the university2 dataset shows very similar results to the previous dataset, namely, a variance value of 0.03 and a standard deviation [44] of 0.18.

**Table 6.** Experimental results of generating SBVR in dataset_university2.

| Question ID | SBVR | Error Due to Wrong Fact Type | Error Due to Wrong Sequencing | Accuracy |
|---|---|---|---|---|
| r1 | - | - | - | - |
| r2 | 7 | - | - | 1.00 |
| r3 | 6 | - | - | 1.00 |
| r4 | 6 | - | - | 1.00 |
| r5 | 6 | - | - | 1.00 |
| r6 | 3 | - | - | 1.00 |
| r7 | 9 | - | - | 1.00 |
| r8 | 3 | - | - | 1.00 |
| r9 | 4 | - | - | 1.00 |
| r10 | 6 | - | - | 1.00 |
| r11 | 5 | - | - | 1.00 |
| r12 | 5 | - | 2 | 0.60 |
| r13 | 3 | - | 1 | 0.67 |
| r14 | 5 | - | 1 | 0.80 |
| r15 | 6 | - | - | 1.00 |
| r16 | 5 | - | - | 1.00 |
| r17 | 5 | - | - | 1.00 |
| r18 | 3 | - | - | 1.00 |
| r19 | 5 | 2 | 1 | 0.40 |
| r20 | 18 | - | - | 1.00 |
| r21 | 7 | - | 1 | 0.86 |
| r22 | 8 | - | - | 1.00 |
| r23 | 5 | - | - | 1.00 |
| r24 | 12 | - | 3 | 0.75 |
| r25 | 5 | 1 | 1 | 0.60 |
| r26 | 7 | - | - | 1.00 |
| r27 | 11 | - | 2 | 0.82 |
| r28 | 4 | 1 | 1 | 0.50 |
| r29 | 7 | - | - | 1.00 |
| r30 | 13 | 2 | 2 | 0.69 |
| r31 | 7 | - | 1 | 0.86 |
| r32 | 3 | - | 1 | 0.67 |
| r33 | 10 | - | - | 1.00 |
| | | | Average | 0.88 |
| | | | Variance | 0.03 |
| | | | Standard deviation | 0.18 |

The SBVR extraction from UML (M2M) result in Skersys, Danenas, and Butleris (2018) has an accuracy value for the original model of 0.70 and for the refactored model of 0.97. The accuracy value of M2M is higher because the SBVR is extracted from UML with an existing and clear structure. It differs from ID2SBVR where the initial data was in the form of natural language with an irregular language structure.

To measure the variance of data distribution related to ID2SBVR accuracy in generating SBVR, the standard deviation of the existing accuracy is determined. When the standard deviation [44] value is low, the extent of the variability in the accuracy values falls within a close range in all processes. Therefore, the ID2SBVR-generated SBVR results are close to each other and do not display any marked deviations. This was the case in both datasets, where the standard deviations of accuracy values of ID2SBVR were 0.15 and 0.18, respectively.

Figure 8 shows the accuracy of dataset_university1 and dataset_university2. ID2SBVR-generated SBVR data are strongly influenced by the fact type extraction results; therefore, differences in accuracy may also occur. Dataset_university2 has an accuracy value 0.06 lower than dataset_university1 because fact type errors and missing fact types occur more commonly than in dataset_university1.

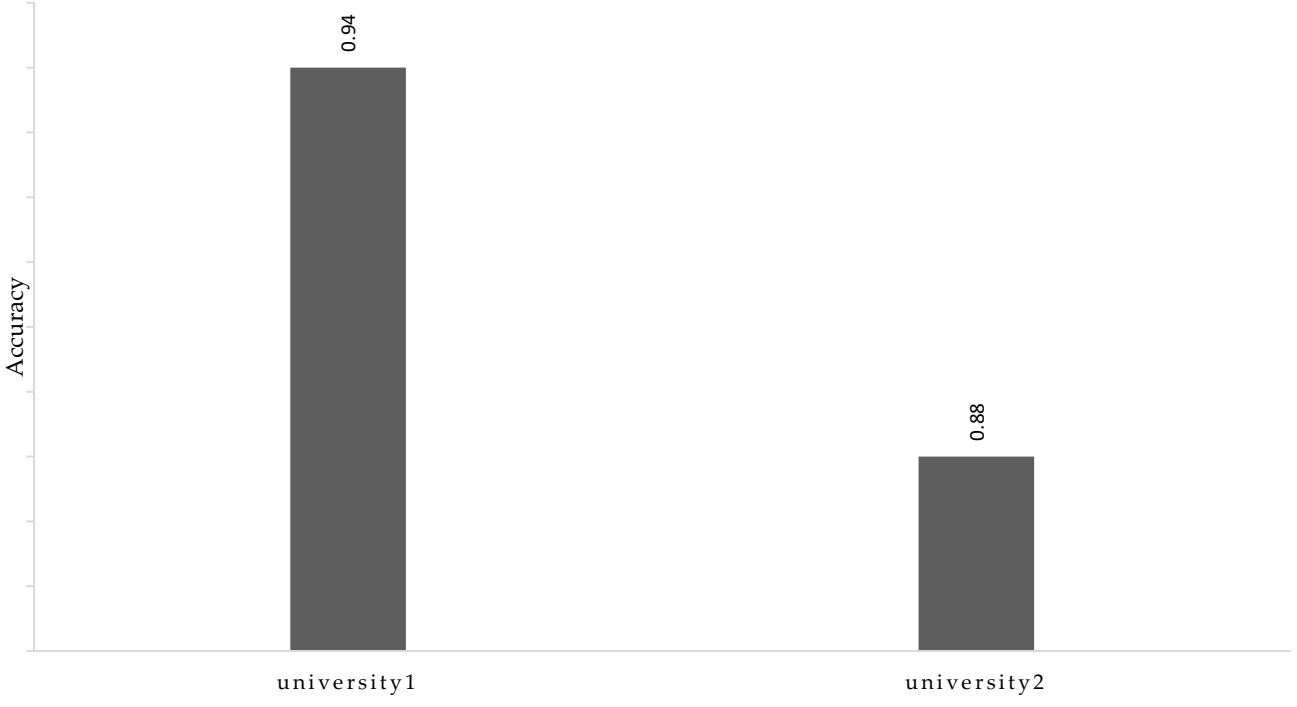

**Figure 8.** Graphical representations of the accuracy of ID2SBVR generating SBVR.

## 5. Threat to Validity

In this study, the threat to validity lies in two phenomena, namely (1) processes that are not mentioned in the interview and (2) processes referred to in interviews that are not sequentially described by the respondent. The validity of the ID2SBVR method has not been tested for the two phenomena above, because the dataset does not contain these phenomena. By design, the ID2SBVR method can be used in different business domains because this method is theoretically independent of domain.

The ID2SBVR method does not depend on the corpus of a particular domain. Although the test results of the ID2SBVR method show very good results in the PPT domain, it has not been tested in other domains. In addition, the dataset that is built is assumed to have a standard grammar. For this reason, in the dataset acquisition process, there is a correction process for meaningless words and for repeated sentences. The complexity of the resulting business processes is not limited. It has fulfilled all existing gateways in BPMN.

## 6. Conclusions

The ID2SBVR presents a new approach for extracting fact types from informal documents. The ID2SBVR allows a business process designer to translate natural language from an interview document into operational rules in SBVR, which in turn can be transformed into BPMN. The novelty of ID2SBVR is that it uses informal documents as a substitute for the formal documents that usually are required by BPMN. The informal documents are the result of an open-ended interview. The data are formed from irregularly structured natural language.

The ID2SBVR succeeds in SBVR operational rule extraction from informal documents on the basis of sentence extraction relevant to SBVR and its sequence. The unstructured data is successfully converted into semi-structured data for use in the pre-processing. The ID2SBVR method translates informal documents that are unstructured into structured ones with a high accuracy value of 0.91. The standard deviation of the ID2SBVR accuracy value in each process is 0.17. The ID2SBVR accuracy value does not show any large data deviations. The ID2SBVR method succeeded in extracting the types of facts including compound, complex, and complex-compound sentences, with an average value of 0.91 for precision and recall, and an almost perfect accuracy of 0.97.

This study has both theoretical and practical implications. Theoretically, this study complements linguistic studies relating to business vocabulary and business rules [9,10,16,25,39] and information retrieval [36,45,46]. Fact types can be extracted based on sequence words or from POS tags. The actor of a fact type with noun phrase can also be extracted. Thus, our research contributes to the extracted user story aspect of what and aspect of who. This new method establishes SBVR using datasets obtained from interview documents. This method succeeds in mining fact type candidates and generating SBVR from informal documents with almost perfect accuracy. The extraction results carried out by ID2SBVR will be used for transformation to a process model using the XML Process Definition Language (XPDL).

The current research has various practical implications, namely: (i) making it easier to understand organizational processes because they are presented in BPMN; (ii) enabling employee assessment of business processes, as BPMN makes them easier to understand; (iii) recommendations for business process improvements within organizations—initial BPMN documentation can be a basic consideration for submitting the re-engineering process; (iv) automatic transformation from SBVR to BPMN; and (v) the method can be used as a basis for comparative analysis between formal BP and actual implementation.

Our future work will focus on extending the method to detect whether there are missing processes, non-sequential processes, or conflicts because of different information obtained from more than one process. The next step would be to perform transformation from SBVR to BPMN by extending the process with another gateway in the BPMN.

**Author Contributions:** Conceptualization, I.T., R.S. and D.S.; methodology, I.T., R.S. and D.S.; validation, I.T.; formal analysis, I.T., R.S. and D.S; investigation, I.T.; data curation, I.T.; writing—original draft preparation, I.T.; writing—review and editing, R.S. and D.S; supervision, R.S. and D.S. All authors have read and agreed to the published version of the manuscript.

**Funding:** This research received no external funding.

**Data Availability Statement:** Data available in a publicly accessible repository: the data presented in this study are openly available in FigShare at https://doi.org/10.6084/m9.figshare.15123879.v1 (accessed on 27 January 2022) and https://doi.org/10.6084/m9.figshare.15123972.v2 (accessed on 27 January 2022).

**Conflicts of Interest:** The authors declare no conflict of interest.

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
