# Peer review of "ID2SBVR: A Method for Extracting Business Vocabulary and Rules from an Informal Document"

_2504-2289, doi:10.3390/bdcc6040119_

Round 1

Reviewer 1 Report

Dear Authors,

First of all I would like to thank you very much for such interesting article.

In the article ID2SBVR: A method for extracting business vocabulary and 2 rules from an informal document the authors introduced a novel approach called informal document to SBVR (ID2SBVR) for extracting operational rules of SBVR from informal documents. ID2SBVR allows a business process designer to translate natural language from an interview document into operational rules in SBVR, which in turn can be trans-636 formed into BPMN. The presented solution is current and relevant for the development of implementation of Process Management in to company management strategy.

The article is clear and relevant for the field of business data analyses area. The structure is in a proper manner. The business implications are clearly indicated. The cited references are recent and selected properly and relevant for the subject of the article. The research procedure may be repeated. The used tables and graphs are easy to interpret and understand. Conclusion are prepared correctly and clearly presented.

Accept in Present Form: The paper is accepted without any further changes.

Author Response

Dear Reviewer 1,

Thank you for your kind appreciation.

Reviewer 2 Report

 The study introduces a new method in doing linguistic analysis, which merits attention. However, it is a bit rough in the overall quality and should need further improvement.

1.      Abstract: some information is too concrete and can be reduced. Thus, only the key points can be better highlighted.

2.      Related work: the authors should highlight the key point in the beginning of each paragraph or it will be quite confusing to get the focal point. Meanwhile, this section lacks a critique of the literature. What the authors will do is not based on the critique or the summary of the related work.

3.      Experiment: are precision, recall, specificity, and accuracy classified by your experiment or which standard? Maybe the too complicated introduction of the method can be shortened whereas the experiment can be more concretely explained.

4.      Implication: the theoretical implication is actually not theoretical, but methodological. The authors should think about what kind of linguistic theory they have enhanced to show the real theoretical contribution. Meanwhile, the practical implication is too short without a detailed explanation on how to use the method to generate related implication.

5.      Conclusion: this section is better arranged before implication and it is better to insert a limitation section. Actually, no method is perfect.

6.      The language needs to be edited, especially in the newly revised part. Its readability is not that good. Many problems also exist in the non-revised section as well.

Author Response

Thank you for your appreciation, comments, and suggestions.

Reviewer 3 Report

The authors note that according to, among others, Mishra & Sureka, there are inconsistencies between Business Process Modeling Notation (BPMN) and Semantics of Business Vocabulary and Business Rules (SBVR). Accordingly, the authors’ proposed value-added proposition is to perform automatic translation of the informal document into fact type and operational rules of SBVR; they entitle their approach ID2SBVR.

The authors should note that their unpacking of acronyms, such as SBVR, is redundant and inconsistent (please refer to lines 57-58; 77). In addition, the clarification of the BPMN/SBVR inconsistencies, expressed in Section 1, is not quite clear, and the value-add of ID2SBVR is not quite illuminated in Section 1 (please refer to lines 67 through 72). While this is somewhat further clarified on lines 118-126, I would suggest providing a sentence or two in Section 1, Even for the clarification on lines 118-126, it is still wanting with regards to explaining the inconsistencies and the value-add of ID2SBVR; for example, does ID2SBVR contribute towards examining subgraph-isomorphism? A few sentences in this regard would be extremely useful.

On line 519, there is an “Error! Reference source not found.” Section 5 seems too abridged, and the reader is left unconvinced. The assertions are high, but the underlying substantiation could be buttressed. By the Conclusion in Section 6, the ID2SBVR distinction (from a technical/mathematical perspective) remains ambiguous.

Author Response

Thank you for your kind appreciation, comments, and suggestions.

Reviewer 4 Report

The paper is very interesting. The topic is current and deserves attention.

The paper is easy to read, the logic is good.

The methodology is clearly presented.

I recommend adding some information to the conclusions related to existing similar methods, or comparisons.

The results part must be presented more clearly!

The bibliography is representative and well chosen.

Author Response

(The authors gave the same response as above.)

Round 2

Reviewer 2 Report

I appreciate the authors' efforts in addressing my comments and overall I believe the quality of the manuscript is acceptable. If possible, I would suggest the authors use professional editing service to improve the language of the manuscript. Then the paper can be better presented. 

Author Response

Thank you for your kind appreciation and suggestion. Please see the attachment.
